# HiLLoC: Lossless Image Compression with Hierarchical Latent Variable Models

**James Townsend**,[*] **Thomas Bird**,[*] **Julius Kunze & David Barber**
Department of Computer Science
University College London
`<firstname>.<surname>@cs.ucl.ac.uk`

## Abstract

We make the following striking observation: fully convolutional VAE models trained on $32\times32$ ImageNet can generalize well, not just to $64\times64$ but also to far larger photographs, with no changes to the model. We use this property, applying fully convolutional models to lossless compression, demonstrating a method to scale the VAE-based 'Bits-Back with ANS' algorithm for lossless compression (Townsend et al., 2019) to large color photographs, and achieving state of the art for compression of full size ImageNet images. We release Craystack, an open source library for convenient prototyping of lossless compression using probabilistic models, along with full implementations of all of our compression results[1].

## 1 Introduction

Bits back coding (Wallace, 1990; Hinton & van Camp, 1993) is a method for performing lossless compression using a latent variable model. In an ideal implementation, the method can achieve an expected message length equal to the variational free energy, often referred to as the negative evidence lower bound (ELBO) of the model. Bits back was first introduced to form a theoretical argument for using the ELBO as an objective function for machine learning (Hinton & van Camp, 1993).

The first implementation of bits back coding (Frey, 1997; Frey & Hinton, 1996) made use of first-in-first-out (FIFO) arithmetic coding (AC) (Witten et al., 1987). However, the implementation did not achieve optimal compression, due to an incompatibility between a FIFO coder and bits back coding, and its use was only demonstrated on a small dataset of $8\times8$ binary images.

Recently, zero-overhead bits back compression with a significantly simpler implementation has been developed by Townsend et al. (2019). This implementation makes use of asymmetric numeral systems (ANS), a last-in-first-out (LIFO) entropy coding scheme (Duda, 2009). The method, known as 'Bits Back with Asymmetric Numeral Systems' (BB-ANS) was demonstrated by compressing the MNIST test set using a variational auto-encoder (VAE) model (Kingma & Welling, 2013; Rezende et al., 2014), achieving a compression rate within $1\%$ of the model ELBO.

More recently, Hoogeboom et al. (2019) and Ho et al. (2019) have proposed flow-based methods for lossless compression, and Kingma et al. (2019) have presented 'Bit-Swap', extending BB-ANS to hierarchical models. In this work we present an alternative method for extending to hierarchical VAEs. This entails the following novel techniques:

1. Direct coding of arbitrary sized images using a fully convolutional model.
2. A vectorized ANS implementation supporting dynamic shape.
3. Dynamic discretization to avoid having to calibrate a static discretization.
4. Initializing the bits back chain using a different codec.

We discuss each of these contributions in detail in Section 3. We call the combination of BB-ANS using a hierarchical latent variable model and the above techniques: 'Hierarchical Latent Lossless

---

[*]Equal contribution.

[1]Available at `https://github.com/hilloc-submission/hilloc`.

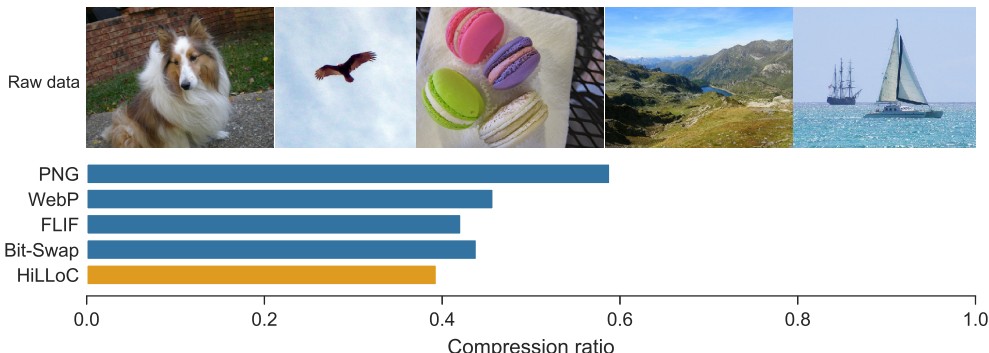

Figure 1: A selection of images from the ImageNet dataset and the compression rates achieved on the dataset by PNG, WebP, FLIF, Bit-Swap and the HiLLoC codec (with ResNet VAE) presented in this work.

Compression' (HiLLoC). In our experiments (Section 4), we demonstrate that HiLLoC can be used to compress color images from the ImageNet test set at rates close to the ELBO, outperforming all of the other codecs which we benchmark. We also demonstrate the speedup, of nearly three orders of magnitude, resulting from vectorization. We release an open source implementation based on 'Craystack', a Python package which we have written for general prototyping of lossless compression with ANS.

## 2 BACKGROUND

In this section we briefly describe the BB-ANS algorithm first introduced by Townsend et al. (2019). We begin by giving a high-level description of the ANS LIFO entropy coder (Duda, 2009), along with a new notation for describing the basic ANS operations. Throughout the rest of the paper we use $\log$ to mean the base two logarithm, usually denoted $\log_2$, and we measure message lengths in bits.

### 2.1 ASYMMETRIC NUMERAL SYSTEMS

As an entropy coder, ANS was designed for compressing sequences of discretely distributed symbols. It achieves a compressed message length equal to the negative log-probability (information content) of the sequence plus an implementation dependent constant, which is usually less than 32 bits. For long sequences, the constant overhead has a negligible contribution to the overall compression rate. Thus, by Shannon's source coding theorem (Shannon, 1948), ANS coding is guaranteed to be near-optimal for long sequences.

There are two basic operations defined by ANS, which we will refer to as 'push' and 'pop'. Push *encodes* a symbol by adding it to an existing message. It has the signature

$$\text{push} : (\text{message}, \text{symbol}) \mapsto \text{message}'. \tag{1}$$

Pop is the inverse of push, and may be used to *decode* a symbol and recover a message identical to that before pushing.

$$\text{pop} : \text{message}' \mapsto (\text{message}, \text{symbol}). \tag{2}$$

When multiple symbols are pushed in sequence, they must be popped using the precise inverse procedure, which means popping the symbols in the opposite order. Hence why ANS is referred to as a last-in-first-out coder, or a *stack*.

The push and pop operations require access to a probabilistic model of symbols, summarized by a probability mass function $p$ over the alphabet of possible symbols. The way that symbols are encoded depends on the model, and pushing a symbol $s$ according to $p$ results in an increase in message length of $\log \frac{1}{p(s)}$. Popping $s$ results in an equal reduction in message length. For details on how the ANS operations are implemented, see Duda (2009).

Note that any model/mass function can be used for the pop operation, i.e. there's no hard restriction to use the distribution that was used to encode the message. In this way, rather than decoding the same

data that was encoded, pop can actually be used to *sample* a symbol from a different distribution. The pop method itself is deterministic, so the source of randomness for the sample comes from the data contained within the message. This sampling operation, which can be inverted by pushing the sample back onto the stack, is essential for bits back coding.

For convenience, we introduce the shorthand notation $s \to p(\cdot)$ for encoding (pushing) a symbol $s$ according to $p$, and $s \leftarrow p(\cdot)$ for decoding (popping).

## 2.2 BITS BACK WITH ANS

Suppose we have a model for data $x$ which involves a *latent variable* $z$. A sender and receiver wish to communicate a sample $x$. They have access to a prior on $z$, denoted $p(z)$, a likelihood $p(x \,|\, z)$ and a (possibly approximate) posterior $q(z \,|\, x)$, but not the marginal distribution $p(x)$. Without access to $p(x)$, sender and receiver cannot directly code $x$ using ANS. However, BB-ANS specifies an indirect way to push and pop $x$. It does not require access to the marginal $p(x)$, but rather uses the prior, conditional, and posterior from the latent variable model.

Table 1(a) shows, in order from the top, the three steps of the BB-ANS pushing procedure which the sender can perform to encode $x$. The 'Variables' column shows the variables known to the sender before each step. 1(b) shows the inverse steps which the receiver can use to pop $x$, with the 'Variables' column showing what is known to the receiver *after* each step. After decoding $x$, the third step of popping, $z \to q(\cdot \,|\, x)$, is necessary to ensure that BB-ANS pop is a precise inverse of push.

Table 1: Indirectly pushing and popping $x$ using BB-ANS. $\to$ and $\leftarrow$ denote pushing and popping respectively. $\Delta L$ denotes the change in message length resulting from each operation. The three steps to push/pop are ordered, starting at the top of the table and descending.

(a) Pushing $x$

| Variables | Operation | $\Delta L$ |
|---|---|---|
| $x$ | $z \leftarrow q(\cdot \,|\, x)$ | $-\log \frac{1}{q(z \,|\, x)}$ |
| $x, z$ | $x \to p(\cdot \,|\, z)$ | $+\log \frac{1}{p(x \,|\, z)}$ |
| $z$ | $z \to p(\cdot)$ | $+\log \frac{1}{p(z)}$ |

(b) Popping $x$

| Operation | Variables | $\Delta L$ |
|---|---|---|
| $z \leftarrow p(\cdot)$ | $z$ | $-\log \frac{1}{p(z)}$ |
| $x \leftarrow p(\cdot \,|\, z)$ | $x, z$ | $-\log \frac{1}{p(x \,|\, z)}$ |
| $z \to q(\cdot \,|\, x)$ | $x$ | $+\log \frac{1}{q(z \,|\, x)}$ |

The change in message length from BB-ANS can easily be derived by adding up the quantities in the $\Delta L$ column of Table 1. For encoding we get

$$\Delta L_{\text{BB-ANS}} = -\log \frac{1}{q(z \,|\, x)} + \log \frac{1}{p(x \,|\, z)} + \log \frac{1}{p(z)} \tag{3}$$

$$= -\log \frac{p(x, z)}{q(z \,|\, x)}. \tag{4}$$

Taking the expectation over $z$ gives the expected message length for a datum $x$

$$\mathcal{L}(x) = -\mathbb{E}_{q(z \,|\, x)} \left[ \log \frac{p(x, z)}{q(z \,|\, x)} \right] \tag{5}$$

which is the negative evidence lower bound (ELBO), also known as the free energy. This is a commonly used training objective for latent variable models. The above equation implies that latent variable models trained using the ELBO are implicitly being trained to minimize the expected message length of lossless compression using BB-ANS.

Note that, as Table 1 shows, the first step of encoding a data point, $x$, using BB-ANS is to, counter-intuitively, *decode* (and thereby sample) a latent $z \leftarrow q(\cdot \,|\, x)$. This requires that there is already a buffer of random data pushed to the ANS coder, which can be popped. This data used to start the encoding process is recovered after the final stage of decoding, hence the name 'bits back'.

If we have multiple samples to compress, then we can use 'chaining', which is essentially repeated application of the procedure in Table 1 (Townsend et al., 2019). In Section 3.4 we describe how we build up an initial buffer of compressed data by using a different codec to code the first images in a sequence.

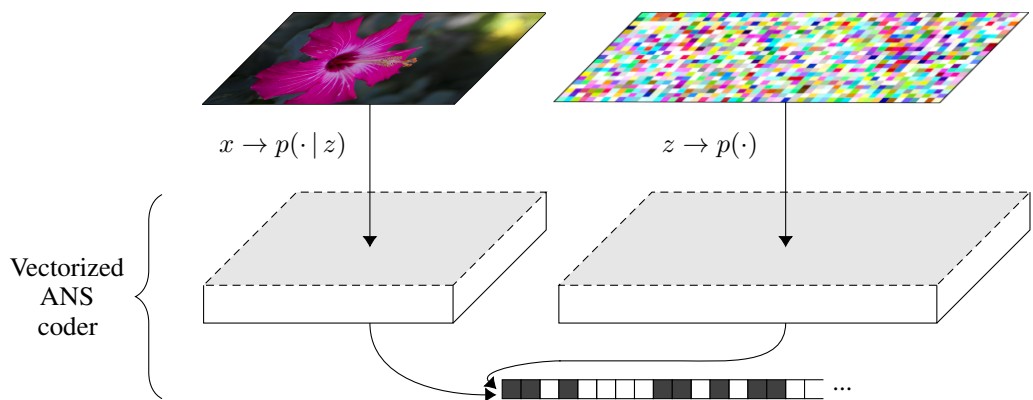

$x \rightarrow p(\cdot \mid z)$ $z \rightarrow p(\cdot)$

Vectorized ANS coder

Figure 2: Visualizing the process of pushing images and latents from a VAE to the vectorized ANS stack with Craystack. The ANS stack head is shaped such that images and latents can be pushed and popped in parallel, without reshaping. Beneath the shaped top of the stack is the flat message stream output by ANS.

## 3 SCALING UP BITS BACK WITH ANS

We now discuss the techniques we introduce to scale up BB-ANS.

### 3.1 FULLY CONVOLUTIONAL MODELS

When all of the layers in the generative and recognition networks of a VAE are either convolutional or elementwise functions (i.e. the VAE has no densely connected layers), then it is possible to evaluate the recognition network on images of any height and width, and similarly to pass latents of any height and width through the generative network to generate an image. Thus, such a VAE can be used as a (probabilistic) model for images of any size.

We exploit this fact, and show empirically in Section 4 that, surprisingly, a fully convolutional VAE trained on $32 \times 32$ images can perform well (in the sense of having a high ELBO) as a model for $64 \times 64$ images as well as far larger images. This in turn corresponds to a good compression rate, and we implement lossless compression of arbitrary sized images by using a VAE in this way.

### 3.2 VECTORIZED LOSSLESS COMPRESSION

The primary computational bottlenecks in the original BB-ANS implementation (Townsend et al., 2019) were loops over data and latent variables occurring in the Python interpreter. We have been able to vectorize these, achieving an implementation which can scale to large ImageNet images. The effect of vectorization on runtime is shown in Figure 4.

A vectorized implementation of ANS was described in Giesen (2014) using SIMD instructions. This works by expanding the size of the ANS stack head, from a scalar to a vector, and interleaving the output/input bit stream. We implement this in our lossless compression library, Craystack, using Numpy. Please refer to the Craystack code and to Giesen (2014) for more detail. We ensure that the compression rate overhead to vectorization is low by using the BitKnit technique described in Giesen (2015), see Appendix D for more detail. Having vectorized, we found that most of the compute time for our compression was spent in neural net inference, whether running on CPU or GPU, which we know to already be reasonably well optimized.

In Craystack, we further generalize the ANS coder using Numpy's n-dimensional array view interface, allowing the stack head to be 'shaped' like an n-dimensional array, or a nested Python data-structure containing arrays. We can then use a shape which fits that of the data that we wish to encode or decode. When coding data according to a VAE we use an ANS stack head shaped into a pair of arrays, matching the shapes of the observation $x$ and the latent $z$. This allows for a straightforward implementation and clarifies the lack of data dependence between certain operations, such as the

$x \to p(\cdot \mid z)$ and $z \to p(\cdot)$ during encoding, which can theoretically be performed concurrently. This vectorized encoding process is visualized in Figure 2.

## 3.3 DISCRETIZATION

It is standard for state of the art latent variable models to use continuous latent variables. Since ANS operates over *discrete* probability distributions, if we wish to use BB-ANS with such models it is necessary to discretize the latent space so that latent samples can be communicated. Townsend et al. (2019) described a *static* discretization scheme for the latents in a simple VAE with a single layer of continuous latent variables, and showed that this discretization has a negligible impact on compression rate. The addition of multiple layers of stochastic variables to a VAE has been shown to improve performance (Kingma et al., 2019; Kingma et al., 2016; Maaløe et al., 2019; Sønderby et al., 2016). Motivated by this, we propose a discretization scheme for hierarchical VAEs with multiple layers of latent variables.

The discretization described in Townsend et al. (2019) is formed by dividing the latent space into intervals of equal mass under the prior $p(z)$. For a hierarchical model, the prior on each layer depends on the previous layers:

$$p(z_{1:L}) = p(z_L) \prod_{l=1}^{L-1} p(z_l \mid z_{l+1:L}). \tag{6}$$

It isn't immediately possible to use the simple static scheme from Townsend et al. (2019), since the marginals $p(z_1), \ldots, p(z_{L-1})$ are not known. Kingma et al. (2019) estimate these marginals by sampling, and create static bins based on the estimates. They demonstrate that this approach can work well. We propose an alternative approach, allowing the discretization to vary with the context of the latents we are trying to code. We refer to our approach as *dynamic discretization*.

In dynamic discretization, instead of discretizing with respect to the marginals of the prior, we discretize according to the *conditionals* in the prior, $p(z_l \mid z_{l+1:L})$. Specifically, for each latent layer $l$, we partition each dimension into intervals which have equal probability mass under the conditional $p(z_l \mid z_{l+1:L})$. This directly generalizes the scheme used in BB-ANS (Townsend et al., 2019).

Dynamic discretization is more straightforward to implement because it doesn't require callibrating the discretization to samples. However it imposes a restriction on model structure, in particular it requires that posterior inference is done *top-down*. This precludes the use of Bit-Swap. In Section 3.3.1 we contrast the model restriction from dynamic discretization with the bottom-up, Markov restriction imposed by Bit-Swap itself.

We give further details about the dynamic discretization implementation we use in Appendix A.

### 3.3.1 MODEL RESTRICTIONS

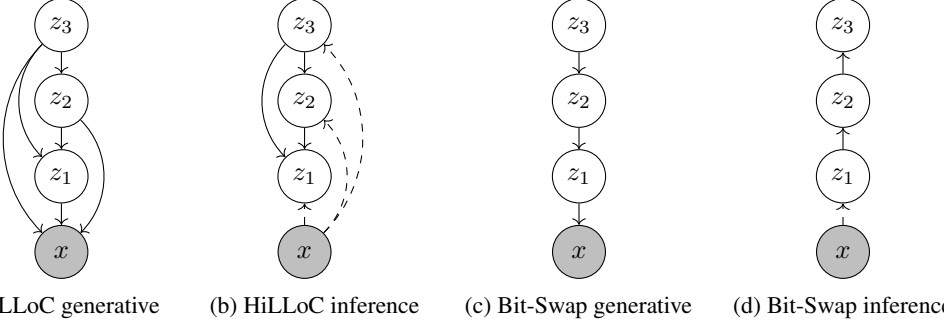

(a) HiLLoC generative     (b) HiLLoC inference     (c) Bit-Swap generative     (d) Bit-Swap inference

Figure 3: Graphical models representing the generative and inference models with HiLLoC and Bit-Swap, both using a 3 layer latent hierarchy. The dashed lines indicate dependence on the fixed observation.

The first stage of BB-ANS encoding is to pop from the posterior, $z_{1:L} \leftarrow q(\cdot \mid x)$. When using dynamic discretization, popping the layer $z_l$ requires knowledge of the discretization used for $z_l$ and

thus of the conditional distribution $p(z_l \mid z_{l+1:L})$. This requires the latents $z_{l+1:L}$ to have already been popped. Because of this, latents in general must be popped (sampled) in 'top-down' order, i.e. $z_L$ first, then $z_{L-1}$ and so on down to $z_1$.

The most general form of posterior for which top-down sampling is possible is

$$q(z_{1:L} \mid x) = q(z_L \mid x) \prod_{l=1}^{L-1} q(z_l \mid z_{l+1:L}, x). \tag{7}$$

This is illustrated, for a hierarchy of depth 3, in Figure 3b. The Bit-Swap technique (Kingma et al., 2019) requires that inference be done bottom up, and that generative and inference models must both be a Markov chain on $z_1, \ldots, z_L$, and thus cannot use skip connections. These constraints are illustrated in Figure 3c,d. Skip connections have been shown to improve model ELBO in very deep models (Sønderby et al., 2016; Maaløe et al., 2019). HiLLoC does not have this constraint, and we do utilize skip connections in our experiments.

### 3.4 STARTING THE BITS BACK CHAIN

As discussed in Section 3.3, our dynamic discretization method precludes the use of Bit-Swap for reducing the one-time cost of starting a BB-ANS chain. We propose instead to use a significantly simpler method to address the high cost of coding a small number of samples with BB-ANS, namely we code the first samples using a different codec. The purpose of this is to build up a sufficiently large buffer of compressed data to permit the first stage of the BB-ANS algorithm - to pop a latent sample from the posterior. In our experiments we use the 'Free Lossless Image Format' (FLIF) (Sneyers & Wuille, 2016) to build up the buffer. We chose this codec because it performed better than other widely used codecs, but in principal any lossless codec could be used.

The amount of previously compressed data required to pop a posterior sample from the ANS stack (and therefore start the BB-ANS chain) is roughly proportional to the size of the image we wish to compress, since in a fully convolutional model the size of the latent space is determined by the image size.

We can exploit this to allow us to obtain a better compression rate than FLIF as quickly as possible. We do so by partitioning the first images we wish to compress with HiLLoC into smaller patches. These patches require a smaller data buffer, and thus we can use the superior HiLLoC coding sooner than if we attempted to compress full images. We find experimentally that, generally, larger patches have a better coding rate than smaller patches. Therefore we increase the size of the image patches being compressed with HiLLoC as more images are compressed and the size of the data buffer grows, until we finally compress full images once the buffer is sufficiently large. For our experiments on compressing full ImageNet images, we compress $32 \times 32$ patches, then $64 \times 64$, then $128 \times 128$ before switching to coding the full size images directly. Note that since our model can compress any shape image, we can compress the edge patches which will have different shape if the patch size does not divide the image dimensions exactly. Using this technique means that our coding rate improves gradually from the FLIF coding rate towards the coding rate achieved by HiLLoC on full images. We compress only 5 ImageNet images using FLIF before we start compressing $32 \times 32$ patches using HiLLoC.

## 4 EXPERIMENTAL RESULTS

Using Craystack, we implement HiLLoC with a ResNet VAE (RVAE) (Kingma et al., 2016). This powerful hierarchical latent variable model achieves ELBOs comparable to state of the art autoregressive models[2]. In all experiments we used an RVAE with 24 stochastic hidden layers. The RVAE utilizes skip connections, which are important to be able to effectively train models with such a deep latent hierarchy. See Appendix E for more details.

We trained the RVAE on the ImageNet 32 training set, then evaluated the RVAE ELBO and HiLLoC compression rate on the ImageNet 32 test set. To test generalization, we also evaluated the ELBO

---

[2]Unlike autoregressive models, for which decoding time scales with number of pixels, and is in practice extremely slow, both encoding and decoding with RVAEs are fast.

and compression rate on the tests sets of ImageNet64, CIFAR10 and full size ImageNet. For full size ImageNet, we used the partitioning method described in 3.4. The results are shown in Table 2.

For HiLLoC the compression rates are for the entire test set, except for full ImageNet, where we use 2000 random images from the test set.

Table 2: Compression performance of HiLLoC with RVAE compared to other codecs. Rates measured in bits/dimension (raw data is 8 bits/dimension). For HiLLoC we display compression rate and theoretical performance (ELBO). All HiLLoC results are obtained from the same model, trained on ImageNet 32.

|  |  | ImageNet 32 | ImageNet 64 | Cifar-10 | ImageNet |
|---|---|---|---|---|---|
| *Generic* | PNG | 6.39 | 5.71 | 5.87 | 4.71 |
|  | WebP | 5.29 | 4.64 | 4.61 | 3.66 |
|  | FLIF | 4.52 | 4.19 | 4.19 | 3.37 |
| *Flow-based* | IDF[3] | 4.18 | 3.90 | 3.34 | - |
|  | IDF generalized[4] | 4.18 | 3.94 | 3.60 | - |
|  | LBB[5] | **3.88** | **3.70** | **3.12** | - |
| *VAE-based* | Bit-Swap | 4.50 | - | 3.82 | 3.51[6] |
|  | HiLLoC | 4.20 | 3.90 | 3.56 | **3.15** |
|  | HiLLoC (ELBO) | (4.18) | (3.89) | (3.55) | (3.14) |

Table 2 shows that HiLLoC achieves competitive compression rates on all benchmarks, and state of the art on full size ImageNet images. The fact that HiLLoC can achieve state of the art compression on ImageNet relative to the baselines, even under a change of distribution, is striking. This provides strong evidence of its efficacy as a general method for lossless compression of natural images. Naively, one might expect a degradation of performance relative to the original test set when changing the test distribution—even more so when the resolution changes. However, in the settings we studied, the *opposite* was true, in that the average per-pixel ELBO (and thus the compressed message length) was *lower* on all other datasets compared to the ImageNet 32 validation set.

In the case of CIFAR, we conjecture that the reason for this is that its images are simpler and contain more redundancy than ImageNet. This theory is backed up by the performance of standard compression algorithms which, as shown in Table 2, also perform better on CIFAR images than they do on ImageNet 32. We find the compression rate improvement on larger images more surprising. We hypothesize that this is because pixels at the edge of an image are harder to model because they have less context to reduce uncertainty. The ratio of edge pixels to interior pixels is lower for larger images, thus we might expect less uncertainty per pixel in a larger image.

To demonstrate the effect of vectorization we timed ANS of single images at different, fixed, sizes, using a fully vectorized and a fully serial implementation. The results are shown in Figure 4, which clearly shows a speedup of nearly three orders of magnitude for all image sizes. We find that the run times for encoding and decoding are roughly linear in the number of pixels, and the time to compress an average sized ImageNet image of $500 \times 374$ pixels (with vectorized ANS) is around 29s on a desktop computer with 6 CPU cores and a GTX 1060 GPU.

## 5 DISCUSSION

Our experiments demonstrate HiLLoC as a bridge between large scale latent variable models and compression. To do this we use simple variants of pre-existing VAE models. Having shown that bits back coding is flexible enough to compress well with large, complex models, we see plenty of work still to be done in searching model structures (i.e. architecture search), optimizing with a trade-off between compression rate, encode/decode time and memory usage. Particularly pertinent for HiLLoC

---

[3]Integer discrete flows, retrieved from Hoogeboom et al. (2019).

[4]Integer discrete flows trained on ImageNet 32. ImageNet 64 images are split into four $32\times32$ patches. Retrieved from Hoogeboom et al. (2019).

[5]Local bits back, retrieved from Ho et al. (2019).

[6] For Bit-Swap, full size ImageNet images were cropped so that their side lengths were multiples of 32.

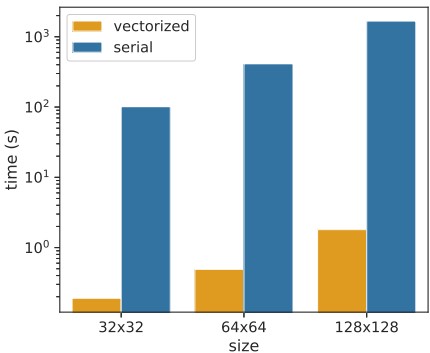

Figure 4: Runtime of vectorized vs. serial ANS implementations. Times were computed on a desktop with 6 CPU cores and a GTX 1060 GPU.

is latent dimensionality, since compute time and memory usage both scale with this. Since the model must be stored/transmitted to use HiLLoC, weight compression is also highly relevant. This is a well-established research area in machine learning (Han et al., 2016; Ullrich et al., 2017).

Our experiments also demonstrated that one can achieve good performance on a dataset of large images by training on smaller images. This result is promising, but future work should be done to discover what the best training datasets are for coding generic images. One question in particular is whether results could be improved by training on larger images and/or images of varying size. We leave this to future work. Another related direction for improvement is batch compression of images of different sizes using masking, analogous to how samples of different length may be processed in batches by recurrent neural nets.

Whilst this work has focused on latent variable models, there is also promise in applying state of the art fully observed auto-regressive models to lossless compression. We look forward to future work investigating the performance of models such as WaveNet (van den Oord et al., 2016) for lossless audio compression as well as PixelCNN++ (Salimans et al., 2017) and the state of the art models in Menick & Kalchbrenner (2019) for images. Sampling speed for these models, and thus decompression, scales with autoregressive sequence length, and can be very slow. This could be a serious limitation, particularly in common applications where encoding is performed once but decoding is performed many times. This effect can be mitigated by using dynamic programming (Le Paine et al., 2016; Ramachandran et al., 2017), and altering model architecture (Reed et al., 2017), but on parallel architectures sampling/decompression is still significantly slower than with VAE models.

On the other hand, fully observed models, as well as the flow based models of Hoogeboom et al. (2019) and Ho et al. (2019), do not require bits back coding, and therefore do not have to pay the one-off cost of starting a chain. Therefore they may be well suited to situations where one or a few i.i.d. samples are to be communicated. Similar to the way that we use FLIF to code the first images for our experiments, one could initially code images using a fully observed model then switch to a faster latent variable model once a stack of bits has been built up.

## 6 CONCLUSION

We presented HiLLoC, an extension of BB-ANS to hierarchical latent variable models, and show that HiLLoC can perform well with large models. We open-sourced our implementation, along with the Craystack package for prototyping lossless compression.

We have also explored generalization of large VAE models, and established that fully convolutional VAEs can generalize well to other datasets, including images of very different size to those they were trained on. We have described how to compress images of arbitrary size with HiLLoC, achieving a compression rate superior to the best available codecs on ImageNet images. We look forward to future work reuniting machine learning and lossless compression.

ACKNOWLEDGMENTS

We thank Paul Rubenstein for the substantial constructive feedback and advice which he gave us. We also thank the anonymous reviewers for their feedback. This work was supported by the Alan Turing Institute under the EPSRC grant EP/N510129/1.

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

## A  REPARAMETERIZING DISCRETIZED LATENTS

After discretizing the latent space, the latent variable at layer $l$ can be treated as simply an index $i_l$ into one of the intervals created by the discretization. As such, we introduce the following notation for pushing and popping according to a discretized version of the posterior.

$$i_l \leftrightarrow Q_l(\cdot \,|\, i_{l+1:L}, x) \tag{8}$$

Where $Q_l(\cdot \,|\, i_{l+1:L}, x)$ is the distribution over the intervals of the discretized latent space for $z_l$, with interval masses equal to their probability under $q(z_l \,|\, \tilde{z}_{l+1:L}, x)$. The discretization is created from splitting the latent space into equal mass intervals under $p(z_l \,|\, \tilde{z}_{l+1:L})$. The mass of a given interval under some distribution is the CDF at the upper bound of the interval minus the CDF at the lower end of the interval. We have used $\tilde{z}$ to indicate that these will be discrete $z_l$ values that are reconstructed from the indices $i_l$. In practise we take $\tilde{z}_l(i_l)$ to be the centre of the interval indexed by $i_l$. It is important to note that the $Q_l$ has an implicit dependence on the previous prior distributions $p(z_k|z_{k+1:L})$ for $k \geq l$, as these prior distributions are required to calculate $\tilde{z}_{l+1:L}$ and the discretization of the latent space.

Since we discretize each latent layer to be intervals of equal mass under the prior, the prior distribution over the indices $i_l$ becomes a uniform distribution over the interval indices, $U(i_l)$, which is not dependent on $i_{\neq l}$. Note that this allows us to push/pop the $i_l$ according to the prior in parallel. The full encoding and decoding procedures with a hierarchical latent model and the dynamic discretization we have described are shown in Table 3. Note that the operations in the two tables are ordered top to bottom.

| Variables | Operation | | | Operation | | Variables |
|---|---|---|---|---|---|---|
| $x$ | $i_L$ | $\leftarrow$ | $Q_L(\cdot \,|\, x)$ | $i_{1:L} \quad \leftarrow \quad U(\cdot)$ | | $i_{1:L}$ |
| $x, i_L$ | $i_{L-1}$ | $\leftarrow$ | $Q_{L-1}(\cdot \,|\, i_L, x)$ | $x \quad \leftarrow \quad p(\cdot \,|\, \tilde{z}_{1:L}(i_{1:L}))$ | | $x, i_{1:L}$ |
| $\vdots$ | $\vdots$ | | | $i_1 \quad \rightarrow \quad Q_1(\cdot \,|\, i_{2:L}, x)$ | | $x, i_{2:L}$ |
| | | | | $i_2 \quad \rightarrow \quad Q_2(\cdot \,|\, i_{3:L}, x)$ | | $x, i_{3:L}$ |
| $x, i_{2:L}$ | $i_1$ | $\leftarrow$ | $Q_1(\cdot \,|\, i_{2:L}, x)$ | | | |
| $x, i_{1:L}$ | $x$ | $\rightarrow$ | $p(\cdot \,|\, \tilde{z}_{1:L}(i_{1:L}))$ | $\vdots$ | | $\vdots$ |
| $i_{1:L}$ | $i_{1:L}$ | $\rightarrow$ | $U(\cdot)$ | $i_L \quad \rightarrow \quad Q_L(\cdot \,|\, x)$ | | $x$ |
| | (a) Encoding | | | | (b) Decoding | |

Table 3: The BB-ANS encoding and decoding operations, in order from the top, for a hierarchical latent model with $l$ layers. The $Q_l$ are posterior distributions over the indices $i_l$ of the discretized latent space for the $l$th latent, $z_l$. The discretization for the $l$th latent is created such that the intervals have equal mass under the prior.

## B  CODEC FOR VARIABLE IMAGE SIZES

Here we describe a codec to compress a set of images of arbitrary size. The encoder now adds the dimensions of the image being coded to the stream of compressed data, such that the decoder knows what shape the image will be before decoding it. Since we are using a vectorized ANS coder, as described in Section 3.2, we resize the top of the coder in between each coding/decoding step such that the size of the top of the coder matches the sizes of the image and latents being coded. The codec is detailed in Table 4.

To make the resizing procedure efficient, we resize via 'folding' the top of the vectorized ANS coder such that we are roughly halving/doubling the number of individual ANS coders each time we fold. This makes the cost of the resize logarithmic with the size difference between the vectorized coder and the targeted size.

Table 4: Codec for an image, $x$, with shape $s$. We code the image via the HiLLoC codec, and the dimensions of the image with the uniform codec, $U$. Since the coder has the same size, $\text{init\_size}$, before and after encoding/decoding, we can use this codec repeatedly to code any number of arbitrary sized images.

| Variables | Operation | | Operation | Variables |
|---|---|---|---|---|
| $x, s$ | $\text{resize\_coder}(s)$ | | $s \leftarrow U(\cdot)$ | $s$ |
| $x, s$ | $x \rightarrow \text{HiLLoC}(\cdot)$ | | $\text{resize\_coder}(s)$ | $s$ |
| $s$ | $\text{resize\_coder}(\text{init\_size})$ | | $x \leftarrow \text{HiLLoC}(\cdot)$ | $x, s$ |
| $s$ | $s \rightarrow U(\cdot)$ | | $\text{resize\_coder}(\text{init\_size})$ | $x, s$ |
| | (a) Encoding | | (b) Decoding | |

## C  COMPRESSION WITH PIXELVAE

To further demonstrate HiLLoC, we implement it with a PixelVAE model. We use a model with two latent layers, although the posterior is fully factorized. The implementation requires nesting an autoregressive codec inside the BB-ANS codec, since the observations and one of the latent layers in PixelVAE have autoregressive generative distributions. Handling this complexity showcases Craystack, which was designed to support this kind of composition. It would also have been prohibitively slow to run on the datasets we compress without the vectorized ANS scheme discussed in Section 3.2.

The achieved compression rate on the entire ImageNet validation set is displayed in Table 5.

The autoregressive component of the PixelVAE generative model leads to an asymmetry between the times required for compression and decompression. Compression with the PixelVAE model is readily parallelizable across pixels, since we already have access to the pixel values we wish to compress and thus also the conditional distributions on each pixel. However, decompression (equivalently, sampling) is not parallelizable across pixels, since we must decompress a pixel value in order to give us access to the conditional distribution on the next pixel. This means the time complexity of decompression is linear in the number of pixels, making it prohibitively slow for most image sizes.

Table 5: The ELBO and compression rate of HiLLoC with PixelVAE, trained to convergence on ImageNet 64, compared to other schemes. All schemes are evaluated on the ImageNet 64 validation set, and measured in bits per pixel-channel.

| | | | | PixelVAE | |
|---|---|---|---|---|---|
| Raw data | PNG | WebP | FLIF | HiLLoC | ELBO |
| 8 | 5.71 | 4.64 | 4.19 | 3.94 | (3.67) |

## D  VECTORIZATION WITHOUT OVERHEADS

To ensure that the compression rate overhead from using vectorization is low, we use a technique from the BitKnit codec (Giesen, 2015). When we reach the end of encoding, we could simply concatenate the integers in the (vector) stack head to form the final output message. However, this is inefficient because the stack head is not uniformly distributed. As discussed in Giesen (2015), elements of the top of the stack have a probability mass roughly

$$p(h) \propto 1/h. \tag{9}$$

Equivalently, the *length* of $h$ is approximately uniformly distributed. More detailed discussion and an empirical demonstration of this is given by Bloom (2014). An efficient way to form the final output message at the end of decoding, is to fold the stack head vector by repeatedly encoding half of it onto the other half, until only a scalar remains, using the above distribution for the encoding. We implement this technique in Craystack and use it for our experiments. The number of (vectorized) encode steps required is logarithmic in the size (i.e. the number of elements) of the stack head.

Some of the overhead from vectorization also comes at the *start* of encoding, when, in existing implementations, the elements of the stack head vector are initialized to copies of a fixed constant. Information from these copies ends up in the message and introduces redundancy which scales with the size of the head. This overhead can be removed by initializing the stack head to a vector of length 1 and then growing the length of the stack head vector gradually as more random data is added to the stack, by *decoding* new stack head vector elements according to the distribution (9).

## E    RESNET VAE ARCHITECTURE

A full description of the RVAE architecture is given in Kingma et al. (2016), and a full implementation can be found in our repository `https://github.com/hilloc-submission/hilloc`, but we give a short description below.

The RVAE is a hierarchical latent model, trained by maximization of the usual evidence lower bound (ELBO) on the log-likelihood:

$$\log p(x) \geq \mathbb{E}_{q(z \mid x)} \left[ \log \frac{p(x, z)}{q(z \mid x)} \right] \tag{10}$$

Take the latent hierarchy to be depth $L$, such that the latents are $z_{1:L}$. There are skip connections in both the generative model, $p(x, z_{1:L})$, and the inference model, $q(z_{1:L} \mid x)$. Due to our requirement of using dynamic discretization, we use a top-down inference model [7]. This means that we can write

$$p(x, z_{1:L}) = p(x \mid z_{1:L}) p(z_L) \prod_{l=1}^{L-1} p(z_l \mid z_{l+1:L}) \tag{11}$$

$$q(z_{1:L} \mid x) = q(z_L \mid x) \prod_{l=1}^{L-1} q(z_l \mid z_{l+1:L}, x) \tag{12}$$

And the ELBO as

$$\log p(x) \geq \ \mathbb{E}_{q(z_{1:L} \mid x)} \left[ \log p(x \mid z_{1:L}) \right] - D_{\mathrm{KL}}(q(z_L \mid x) \,\|\, p(z_L)) \tag{13}$$

$$- \sum_{l=1}^{L-1} \mathbb{E}_{q(z_{l+1:L} \mid x)} \left[ D_{\mathrm{KL}}(q(z_l \mid z_{l+1:L}, x) \,\|\, p(z_l \mid z_{l+1:L})) \right] \tag{14}$$

Where $D_{\mathrm{KL}}$ is the KL divergence. As in Kingma et al. (2016), the KL terms are individually clamped as $\max(D_{\mathrm{KL}}, \lambda)$, where $\lambda$ is some constant. This is an optimization technique known as *free bits*, and aims to prevent latent layers in the hierarchy collapsing such that the posterior is equal to the prior.

Each layer in the hierarchy consists of a ResNet block with two sets of activations. One set of activations are calculated bottom-up (in the direction of $x$ to $z_L$), and the other are calculated top-down. The bottom-up activations are used only within $q(z_{1:L} \mid x)$, whereas the top-down activations are used by both $q(z_{1:L} \mid x)$ and $p(x, z_{1:L})$. Every conditional distribution on a latent $z_l$ is parameterized as a diagonal Gaussian distribution, with mean and covariance a function of the activations within the ResNet block, and the conditional distribution on $x$ is parameterized by a discretized logistic distribution. Given activations for previous ResNet blocks, the activations at the following ResNet block are a combination of stochastic and deterministic features of the previous latent layer, as well as from skip connections directly passing the previous activations. The features are calculated by convolutions.

Note also that all latent layers are the same shape. Since we retained the default hyperparameters from the original implementation, each latent layer has 32 feature maps and spatial dimensions half those of the input (e.g. $\frac{h}{2} \times \frac{w}{2}$ for input of shape $h \times w$).

---

[7]Note that in Kingma et al. (2016), this is referred to as bidirectional inference.

