# OpenReview forum: "HiLLoC: lossless image compression with hierarchical latent variable models"
_ICLR.cc/2020/Conference — Accept (Poster)_

### Official Review · AnonReviewer3 · 2019-10-23
**Official Blind Review #3**

**Rating:** 8

**Review:**

Summary:
This paper focuses on lossless source compression with bits back coding for hierarchical fully convolutional VAEs. The focus/contribution is three-fold: 1. Improve the compression rate performance by adapting the discretization of latent space required for the entropy coder ANS. The newly proposed discretization scheme allows for a dependency structure that is not restricted to a Markov chain structure in the encoder model q(z|x) and in the generative part of the model p(x,z). This is in contrast with bit-swap[1], which requires a markov chain structure. The dependency structure that is allowed in the proposed method is widely known to perform better than a markov chain structure, which can explain why it improves significantly over Bit-swap [1] (another hierarchical VAE compression algorithm that uses bits back coding.) 2. Increasing compression speed by implementing a vectorized version of ANS, and heaving an ANS head in the shape of a pair of arrays matching that of the latent variable and the observed variable. The latter allows for simultaneous encoding of the latent with the prior distribution and the image with the decoder distribution. 3. Showing that a model trained on a low-resolution imagenet 32 dataset can generalize its compression capabilities to higher resolution datasets with convincing results.

Decision: Accept.
This paper is clearly written, makes clear claims and supports these claims with convincing experiments. The contributions are of practical use and I expect future work to benefit from this paper.


Supporting arguments for decision:
The paper is well motivated; off the shelf compression algorithms such as PNG are also not trained on every dataset separately, and cross-dataset generalization is important if this model should be used in practice for many different images from different datasets and of different resolutions.

The paper clearly supports the main claims. It improves upon the previous bits-back coding-based hierarchical VAE [1]. The only hypothesis that is not checked is the one that hypothesizes that the lower bpd for higher resolution images is due to the lower ratio of edge pixels versus non-edge pixels, but this is not a dealbreaker from my point of view.

I would like the authors to revise their statement of state of the art compression performance on page 7 directly below table 2. “The fact that HiLLoC achieves state of the art compression rates relative to the baselines even under a change of distribution is striking, and provides strong evidence of its efficacy as a general method for lossless compression of natural images”. This is sentence should be made more nuanced as the proposed model only improves on Bit-Swap, but is still significantly outperformed by Local bits back coding (LBB [2]), and in the case of cifar-10 also by integer discrete flows (IDF [3]). On the other hand, it would be useful to still state that LBB is trained on every dataset separately, as well as IDF. Note also that in [3], a model that is trained on Imagenet32 and evaluated on the other datasets is also reported (see table 1 in [3]). It would be beneficial for the author to include the scores of this model, as the proposed method seems to perform slightly better at generalizing to new datasets.

Because of the buffer of initial bits required by bit-back coding, the compression/decompression of several data points has to be sequential if one wants to amortize this cost over several data points. Compression methods that don’t rely on bits-back coding, such as IDF [3], do not have this issue and can compress/decompress data points in parallel. Since this influences the practical usability of the model, it would be transparent to mention this.

My final main question is on the equivalence of evaluation methods of Bit-Swap and Hilloc on imagenet. The Bit-Swap paper states: “For MNIST, CIFAR-10 and Imagenet (32 × 32) we report the bitrates, shown in Table 5, as a result of compressing 100 datapoints in sequence (averaged over 100 experiments)...”. This means that Bit-Swap is not evaluated on the full test set of Imagenet 32 (as this contains 50000 images), as opposed to Hilloc. Do the authors think this is a problem?
Furthermore, in the case of “full” Imagenet, Bit-swap uses a subset of 100 images for evaluation and crops them to a multiple of 32 pixels in height and width, so that bit-swap can compress patches and the result is the average of patches for on image. Hilloc appears to take 500 random images and does not state anything about cropping. Could the authors comment on this?



Additional feedback to improve paper (not part of decision assessment):
- In the introduction, first paragraph: “ the method can achieve an expected message length equal to the variational free energy, often referred to as the evidence lower bound (ELBO) of the model. “ → “ the method can achieve an expected message length equal to the variational free energy, often referred to as the negative evidence lower bound (ELBO) of the model. “
- Section 3.2, last paragraph: It is not clear if in practice the latent and image are actually encoded in parallel as the author states that this is “in theory” possible.
- Page 4: “... we found that most of the compute time for our compression was spent in neural net inference, …” I assume you mean “inference” in any part of the encoder or decoder, and not specifically approximate inference of the encoder network. Perhaps clarify this to avoid confusion?
- Section 4: When referring to the ResnetVAE by Kingma et al, it would be appropriate to also cite [4], as this is very similar to resnetVAE’s and was released earlier.



[1] F. H. Kingma, P. Abbeel, and J. Ho. Bit-Swap: recursive bits-back coding for lossless compression with hierarchical latent variables. In International Conference on Machine Learning (ICML), 2019.
[2] Jonathan Ho, Evan Lohn, and Pieter Abbeel. Compression with Flows via Local Bits-Back Coding. arXiv e-prints, 2019.
[3] Emiel Hoogeboom, Jorn W. T. Peters, Rianne van den Berg, and Max Welling. Integer Discrete Flows and Lossless Compression. arXiv e-prints, 2019.
[4] C. K. Sønderby, T. Raiko, L. Maaløe, S. K. Sønderby, and O. Winther. Ladder variational autoencoders. In Advances in Neural Information Processing Systems (NIPS), 2016.


**Experience Assessment:**

I have published one or two papers in this area.

**Review Assessment: Checking Correctness Of Derivations And Theory:**

N/A

**Review Assessment: Checking Correctness Of Experiments:**

I carefully checked the experiments.

**Review Assessment: Thoroughness In Paper Reading:**

I read the paper at least twice and used my best judgement in assessing the paper.

---

> ### Author Response · Authors · 2019-11-13
> **Response to Review #3**
>
> Thank you for your review. To address the points you raised:
>
> >I would like the authors to revise their statement of state of the art compression performance on page 7 directly below table 2. ... It would be beneficial for the author to include the scores of [IDF generalization].
>
> We have revised the statement below Table 2 to more accurately reflect the data in the table. We have also added the results for IDF generalization to the table.
>
> >Because of the buffer of initial bits required by bit-back coding, the compression/decompression of several data points has to be sequential if one wants to amortize this cost over several data points. Compression methods that don’t rely on bits-back coding, such as IDF [3], do not have this issue and can compress/decompress data points in parallel. Since this influences the practical usability of the model, it would be transparent to mention this.
>
> We have added mention of these models to the last paragraph of the Discussion section, where we felt this point fitted.
>
> >My final main question is on the equivalence of evaluation methods of Bit-Swap and Hilloc on imagenet. The Bit-Swap paper states: “For MNIST, CIFAR-10 and Imagenet (32 × 32) we report the bitrates, shown in Table 5, as a result of compressing 100 datapoints in sequence (averaged over 100 experiments)...”. This means that Bit-Swap is not evaluated on the full test set of Imagenet 32 (as this contains 50000 images), as opposed to Hilloc. Do the authors think this is a problem?
>
> This implies that the Bit-Swap results may be noisier than ours. They also give ‘average net bitrate’ values in tables 2-4, which are close to the values in their table 5. We presume that the error bounds that they give are 2 standard deviations, from the empirical distribution of the ‘100 experiments’ they ran. We think it’s likely that the figures they give are accurate enough to reasonably compare to ours.
>
> >Furthermore, in the case of “full” Imagenet, Bit-swap uses a subset of 100 images for evaluation and crops them to a multiple of 32 pixels in height and width, so that bit-swap can compress patches and the result is the average of patches for on image. Hilloc appears to take 500 random images and does not state anything about cropping. Could the authors comment on this?
>
> We have updated our results after benchmarking on a larger subset of 2000 (not 500) images, and have updated the paper to reflect this. The Bit-Swap result here may be affected by noise due to the smaller scale of their experiment, however again we have assumed that it is accurate enough for comparison. The BitSwap images are indeed cropped so that the side lengths are multiples of 32. For HiLLoC this was not necessary. We think the comparison still makes sense even with this slight difference, and we have added a footnote to explain the difference between our full size ImageNet experiment and the one in Bit-Swap.

---

> > ### Comment · AnonReviewer3 · 2019-11-15
> > **Response to rebuttal**
> >
> > I am satisfied with the author's rebuttal, and will keep my rating at "Accept".

---

### Official Review · AnonReviewer1 · 2019-10-24
**Official Blind Review #1**

**Rating:** 6

**Review:**

The authors propose a method for lossless image compression based on using
fully convolutional VAE models. These models are shown to generalize well when
they are trained on small images (e.g. 32x32 and 64x64) and then applied to
much larger images. The method is based on a fully vectorized implementation of
bits back with asymetric numeral systems coding which is much faster than
previous non-vertorized implementations. An improvement with respect to similar
methods is to use a dynamic discretization of the latent variables which avoids
having to callibrate a static discretization (as in previous methods).
Finally, the authors initialize the bis back process with information about a
few initial images which are coded using a different codec.  The experiments
performed illustrate the gains of the method in terms of compression ratio and
speed.

Clarity:

The paper is extremelly well writen and it is very easy to read. The athors
indicate that they will release open-source code to implement all their
results, which is very wellcome to improve reproducibility. However, I have to
say that the part describing the vectorized implementation of their method was
rather confusing and the paper could benefit a lot from clarifying this part.

Quality:

The experiments performed are sound and illustrate the gains produced by their
method (although they do not achieve state of the art results). In particular,
the experiments show the speed up gain by the proposed vectorization and the gains
produced by the dynamic discretization. The experiments also show how the methods
trained on smaller images generalize well to larger images.

Novelty:

The proposed approach is novel up to my knowledge. Although the methodological
innovations are not that advanced, the vectorization in the specific
application considered is novel, as well as the dynamic discretization.

Significance:

The proposed contributions are significant in my opinion. The vectorization
approach can be very useful in practice and the dynamic discretization can also
be useful as shown by the experiments. One criticism could be that the authors
do not achieve state of the art results, but I consider this a minor thing.

**Experience Assessment:**

I have published one or two papers in this area.

**Review Assessment: Checking Correctness Of Derivations And Theory:**

I assessed the sensibility of the derivations and theory.

**Review Assessment: Checking Correctness Of Experiments:**

I assessed the sensibility of the experiments.

**Review Assessment: Thoroughness In Paper Reading:**

I read the paper at least twice and used my best judgement in assessing the paper.

---

> ### Author Response · Authors · 2019-11-13
> **Response to Review #1**
>
> Thank you for your review. We address the following point:
>
> > However, I have to say that the part describing the vectorized implementation of their method was rather confusing and the paper could benefit a lot from clarifying this part.
>
> It’s difficult to give a proper description of this without going into a lot more detail about ANS implementation. To aid readers who are confused and/or curious, we’ve added a recommendation, in the second paragraph of Section 3.2, to refer to our code and to Giesen (2015) for more detail.

---

### Official Review · AnonReviewer2 · 2019-10-25
**Official Blind Review #2**

**Rating:** 6

**Review:**

This paper proposes a method for lossless image compression consisting of a VAE and using a bits-back version of ANS. The results are very impressive on a ImageNet (but maybe not so impressive on the other benchmarks). The authors also discuss how to speed up inference and present some frightening runtime numbers for the serial method, and some better numbers for the vectorized version, though they're nowhere close to being practical.

I think this paper should be accepted. It has a better description of the BB ANS algorithm than I have read before, and it's a truly interesting direction for the field, despite the lack of immediate applicability.

If we are to accept this paper, I suggest the authors put a full description of the neural network used (it's barely mentioned). I think the authors also need to disclose how long it took to compress an average imagenet image (looking at the runtime numbers for 128x128 pixels is scary, but at least we'd get a better picture on the feasability).

Overall, due to the fact that the authors pledge to open source the framework, I think some of the details will be found in the code, once released. I think this is an important step because there are so many details in this paper that one cannot reasonably reproduce the work by simply reading the text of this paper.

**Experience Assessment:**

I do not know much about this area.

**Review Assessment: Checking Correctness Of Derivations And Theory:**

I did not assess the derivations or theory.

**Review Assessment: Checking Correctness Of Experiments:**

I carefully checked the experiments.

**Review Assessment: Thoroughness In Paper Reading:**

I read the paper at least twice and used my best judgement in assessing the paper.

---

> ### Author Response · Authors · 2019-11-13
> **Response to Review #2**
>
> Thank you for your review. To address the points you raised:
>
> > ... put a full description of the neural network used
>
> We have now added a detailed description of the VAE that we used, in Appendix E.
>
> > the authors also need to disclose how long it took to compress an average ImageNet image
>
> We’ve found the encode/decode times are roughly linear in the number of pixels, and you can extrapolate from the graph. To demonstrate this point, we’ve timed compressing ImageNet images with dimension 500x374, which is slightly over the average size. The compression takes 29s. We agree that it's important to disclose this and we’ve added this information to the paper near the end of Section 4. We also agree that these times are slow, and mean that the method is not yet practical. However, we have improved significantly over existing work, and we see plenty of scope for further optimizing the runtime of the algorithm. In particular, quite a lot of code is still running in the Python interpreter, which could be written in another, compiled language. Also the hierarchical VAE that we used was mainly chosen to demonstrate the scalability of the method and to ensure an excellent compression rate, and not for its practicality. A smaller model would almost certainly be more appropriate in the long run, and distillation could be used to minimize runtime whilst maintaining similar compression performance.

---

### Decision · Program_Chairs · 2019-12-19

**Decision:**

Accept (Poster)

**Comment:**

The paper proposes a lossless image compression consisting of a hierarchical VAE and using a bits-back version of ANS. Compared to previous work, the paper (i) improves the compression rate performance by adapting the discretization of latent space required for the entropy coder ANS (ii) increases compression speed by implementing a vectorized version of ANS (iii) shows that a model trained on a low-resolution imagenet 32 dataset can generalize its compression capabilities to higher resolution.

The authors addressed properly reviewers' concerns. Main critics which remain are (i) the method is not practical yet (long compression time) (ii) results are not state of the art - but the contribution is nevertheless solid.